# A Comprehensive Review of Rosmarinic Acid: From Phytochemistry to Pharmacology and Its New Insight

**DOI:** 10.3390/molecules27103292

**Published:** 2022-05-20

**Authors:** Huaquan Guan, Wenbin Luo, Beihua Bao, Yudan Cao, Fangfang Cheng, Sheng Yu, Qiaoling Fan, Li Zhang, Qinan Wu, Mingqiu Shan

**Affiliations:** 1School of Chinese Medicine, Nanjing University of Chinese Medicine, Nanjing 210023, China; guanhuaquan@njucm.edu.cn (H.G.); luowb2005@163.com (W.L.); 290069@njucm.edu.cn (Q.F.); 2Jiangsu Collaborative Innovation Center of Chinese Medicinal Resources Industrialization, Nanjing University of Chinese Medicine, Nanjing 210023, China; baobh@njucm.edu.cn (B.B.); raindc@163.com (Y.C.); ffcheng@njucm.edu.cn (F.C.); yusheng1219@163.com (S.Y.); zhangli@njucm.edu.cn (L.Z.); qnwyjs@163.com (Q.W.); 3School of Pharmacy, Nanjing University of Chinese Medicine, Nanjing 210023, China

**Keywords:** rosmarinic acid, natural product, pharmacokinetics, pharmacology, phytochemistry

## Abstract

Polyphenolic acids are the widely occurring natural products in almost each herbal plant, among which rosmarinic acid (RA, C_18_H_16_O_8_) is well-known, and is present in over 160 species belonging to many families, especially the Lamiaceae. Aside from this herbal ingredient, dozens of its natural derivatives have also been isolated and characterized from many natural plants. In recent years, with the increasing focus on the natural products as alternative treatments, a large number of pharmacological studies have been carried out to demonstrate the various biological activities of RA such as anti-inflammation, anti-oxidation, anti-diabetes, anti-virus, anti-tumor, neuroprotection, hepatoprotection, etc. In addition, investigations concerning its biosynthesis, extraction, analysis, clinical applications, and pharmacokinetics have also been performed. Although many achievements have been made in various research aspects, there still exist some problems or issues to be answered, especially its toxicity and bioavailability. Thus, we hope that in the case of natural products, the present review can not only provide a comprehensive understanding on RA covering its miscellaneous research fields, but also highlight some of the present issues and future perspectives worth investigating later, in order to help us utilize this polyphenolic acid more efficiently, widely, and safely.

## 1. Introduction

In recent years, with in-depth studies of the plants, natural products have increasingly attracted the attention of researchers in many fields. Rosmarinic acid (RA, C_18_H_16_O_8_, Figure 1) is an interesting and well-known representative. Regarding its chemical structure, this naturally-occurring phenol acid is considered as an ester, the esterification product of a caffeic acid and a 3,4-dihydroxyphenyl lactic acid. To our knowledge, it was from *Rosmarinus officinalis* L. that RA was first isolated and identified by two Italian scientists, Scarpati and Oriente, and was named according to the name of this herbal plant [1]. From then on, RA has been successively found in more than 160 plants belonging to Lamiaceae, Boraginaceae, Apiaceae, etc. It has also been investigated for its miscellaneous pharmacological activities including anti-oxidative activity, anti-inflammatory activity, anti-viral activity, anti-diabetic activity, anti-tumor activity, and neuroprotective activity in many in vitro and in vivo studies. Due to its higher content and similar bioactivity to phytomedicines, RA is employed as the quality indicative component for them including *Perilla frutescens* (L.) Britt fruits and stems, *Prunella vulgaris* L. spikes, and *Sarcandra glabra* (Thunb.) Nakai whole plants in the Chinese Pharmacopoeia, *Melissa officinalis* L. leaves and *Eclipta prostrata* (L.) L. aerial parts in the European Pharmacopoeia, and *Rosmarinus o**fficinalis* L. leaves in the United States Pharmacopoeia.

As a variety of studies have been performed and some achievements have been made recently, some reviews of RA have been published [2,3,4,5,6,7,8,9]. However, these articles have mainly focused on pharmacological studies such as its neuroprotective, anti-diabetic, anticancer, and anti-inflammatory potential, which seemed to be a little simplex. Thus, there is still a lack of a comprehensive review to provide a full-scale understanding of this polyphenol acid. In the present study, we used some mainstream bibliographic databases and search engines such as the Web of Science, PubMed, Chinese National Knowledge Infrastructure (CNKI), and Google Scholar to collect a large number of the research literature and to sum up the interesting progress. Except for “rosmarinic acid” as the keyword, some other characteristic words were also employed including ”isolated” for phytochemistry, “positive drug” and “model” for pharmacology, and “pharmacokinetic” and “LC-MS” for pharmacokinetics. Aside from a summary, we also explored some oof the interesting and attractive research issues, which are proposed here and are believed to be the potential hotspots in the future.

## 2. Sources and Biosynthesis in the Plants

To our knowledge, RA has been found and isolated as a monomeric component from a total of 162 plants, which are listed in Table 1. It is obvious that Lamiaceae is the largest family, containing 104 plants among them. As far as the genus containing RA is concerned, Salvia is the largest, with 20 plants including *S. absconditiflora* Greuter & Burdet, *S. deserta* Schang, *S. grandifolia*, *S. miltiorrhiza* Bunge, *S. plebeia* and *S. przewalskii* Maxim, etc. With respect to chemotaxonomy, the existence of RA could provide some taxonomic basis at the level of the subfamily. According to the database of the European and Mediterranean Plant Protection Organization, among the 104 plants of Lamiaceae, 93 species come from Nepetoideae and 10 species come from Lamioideae [10]. It is obvious that RA is a characteristic natural product distinguishing Nepetoideae and other subfamilies in Lamiaceae. However, to carry out a taxonomic study in Lamiaceae more accurately, it is impossible to depend solely on RA. Characteristic terpenoids should play the same important roles.

Many of these 154 plants have been used as the sources of traditional Chinese medicinal materials for a long time such as *Perilla frutescens* (L.) Britt, *Prunella vulgaris* L., *Salvia miltiorrhiza* Bunge, *Sarcandra glabra* (Thunb.) Nakai, *Schizonepeta tenuifolia* Briquet, etc. Some others also serve as folk medicinal plants in many countries and regions such as *Cordia bicolor*, *Cordia dentate*, *Cordia megalantha*, *Hyptis atrorubens* Poit., and *Hyptis verticillata* Jacq. in Central America and the Caribbean; *Micromeria myrtifolia* Boiss. & Hohen, *Salvia palaestina* Bentham and *Sanicula europaea* L. in Turkey; *Baccharis chilco*, *Hyptis capitata* Jacq., and *Lepechinia meyenii* (Walp.) Epling in South America; *Ipomoea turpethum* (L.) R.Br., *Thunbergia laurifolia Lindl,* and *Thymus serpyllum* in South Asia, etc.

It has been shown that in the plants, two amino acids are separately involved in the RA biosynthesis pathways (Figure 2). In the first pathway, *L*-phenylalanine is orderly transformed to cinnamic acid, 4-coumaric acid, and 4-coumaroyl-CoA by phenylalanine ammonia-lyase, cinnamate 4-hydroxylase, and 4-hydroxycinnamate-CoA ligase, respectively. In the second pathway, *L*-tyrosine, the precursor, is first transformed to 4-hydroxyphenylpyruvic acid by tyrosine aminotransferase, and then to 4-hydroxyphenyllactic acid by the hydroxyphenylpyruvate reductase. The products of the two biosynthesis ways, both 4-coumaroyl-CoA and 4-hydroxyphenyllactic acid, could be finally converted into RA by the rosmarinic acid synthase and cytochrome P450 monooxygenase associated with the cytochrome P450 reductase [181,182,183,184]. Therefore, it can be easily concluded that rosmarinic acid synthase is a key control point for both of the above two synthesis pathways. As a member of the BAHD acyltransferase family, it was acidic stable and its molecular mass was tested between 36 kD and 59 kD. It was also characterized with the random curl and α-helix, containing neither signal peptides nor leading peptides in the secondary structure [185,186].

Except for the medicinal parts of the herbal plants as natural RA isolation sources, some non-medicinal parts also serve, where *Salvia miltiorrhiza* aerial parts are a good example. When the roots and rhizomes are harvested, a mass of aerial parts will be thrown and wasted. It was reported that the RA content could reach above 20 mg/g in the aerial parts [187] and Shi et al. successfully isolated it from *Salvia miltiorrhiza* leaves [144]. *Foeniculum*
*Vulgare* Mill. is an aromatic plant and is often extracted by distillation for its volatile oil. However, the resultant residue is considered as a waste, in which many antioxidant components exist. Parejo et al. reported the isolation of RA with seven other phytochemicals in 2004 [58]. Regarding the beach waste, *Zostera noltii* and *Zostera marina* have also been used as the sources of RA isolation [179,180]. In recent years, due to the increasing price of herbal plants, it is of great interest to look for new sources for the isolation of natural products such as RA. Non-medicinal parts and other biowaste, considered as useless and burdensome in the past, have now attracted more and more research. Therefore, the isolation of RA from these new sources would achieve the aims of saving resources, protecting the environment, realizing the efficient use and recycling of resources, and promoting the development of the industrial economy.

## 3. Extraction from Plants

As a naturally-occurring polyphenolic acid, RA has often been obtained from plants by different extraction methods including vibration [188], maceration with continuous stirring [189], heat reflux [189], and Soxhlet solvent extraction [190]. In these traditional extraction methods, the solvent is often a key factor responsible for the RA yield. In a study that extracted RA from *Dracocephalum moldavica* L. aerial parts, *n*-butanol was investigated as the best solvent when the Soxhlet solvent extraction method was used. Compared to the extraction efficiency of *n*-butanol (114.54 ± 24.70 mg/g), those of other solvents were 78.43%, 8.96%, 20.84%, and 8.26% for methanol (89.83 ± 1.38 mg/g), ethyl acetate (10.26 ± 1.29 mg/g), acetonitrile (23.87 ± 0.50 mg/g), and water (9.46 ± 0.07 mg/g), respectively [190].

During the past decades, due to their simpler operation, lower time consumption, and simultaneous preparation of more samples, some novel extraction methods have been utilized for RA extraction such as ultrasound-assisted extraction [191], microwave-assisted extraction [192], enzyme-assisted extraction [193], and pressurized-liquid extraction [194]. In a comparison study of the extraction methods used for the leaves of six plants, to obtain the highest extraction efficiency of RA, the optimal extraction parameters of different methods were as follows: 120 min at 25 °C for maceration with stirring extraction (MACs), 15 min at boiling point for heat reflux extraction (HRE), and 5 min at 50 °C and 80 °C for microwave-assisted extraction (MAE). A mixed solvent (CH_3_CH_2_OH–H_2_O–HCl, 70:29:1, *v*/*v*/*v*) was also proven to be the best for each method. In light of the RA yield, MACs was the most appropriate for *Melissa officinalis* L. (30.0 ± 0.2 mg/g), *Mentha piperita* (16.2 ± 0.6 mg/g), *Rosmarinus officinalis* L. (9.2 ± 0.2 mg/g), and *Salvia officinalis* L. (19.6 ± 0.3 mg/g) while HRE was the best for *Thymus vulgaris* L. (15.3 ± 1.2 mg/g) and *Origanum vulgare* L. (40.1 ± 1.0 mg/g) [195]. In addition, characterized with lower melting points, lower cost, lower vapor pressure, and reproducibility, ionic liquid has become an efficient and environmentally-friendly extraction solvent alternative to the conventional ones. In an ultrasound-assisted extraction study of RA from *Rosmarinus officinalis* leaves, 1-octyl-3-methylimidazolium bromide ([C_8_mim]Br) was selected as the solvent due to its high extraction efficiency. After the optimization of the extraction factors with response surface methodology, the optimal conditions included 2 h for the soaking time, 30 min for the ultrasound time, 220 W for the ultrasound power, and 1:20 for the solid–liquid ratio, under which the extraction efficiency of RA could reach to 98.91% [196]. In another study of microwave-assisted extraction for RA from *Rosmarinus officinalis* leaves, [C_8_mim]Br was also used as the solvent with 700 W for the irradiation power, 15 min for the irradiation time, and 1:12 for the solid–liquid ratio. This method exhibited a considerable RA yield (3.97 mg/g) [197].

## 4. Natural Derivatives in Plants

From a variety of natural plants, a large number of RA derivatives have been found and isolated, which often simultaneously exist in the same plant with RA in most cases. Appendix A demonstrates their chemical structures.

Among these derivatives, the alkyl rosmarinates are the simplest ones in terms of their chemical structures. Due to the C8′-carboxyl group, RA can combine with some alcohol compounds to obtain some esters in the plants such as rosmarinic acid methyl ester, ethyl ester, *n*-propyl ester, and *n*-butyl ester. These alkyl rosmarinates have demonstrated anti-oxidative, anti-inflammatory, anti-allergic, anti-bacterial, anti-cardiovascular disease, and other activities [198,199,200,201,202,203]. In addition, 3-*O*-methyl rosmarinic acid, 3-*O*-caffeoyl rosmarinic acid, 3′-*O*-methyl rosmarinic acid, 4′-*O*-methyl rosmarinic acid (shimobashiric acid B), and 3, 3′-*O*-diethyl rosmarinic acid are the natural products of RA substituted by a methyl, ethyl, and even caffeoyl groups on the C3-, C3′-, and C4′-hydroxyl groups. As a polyphenolic acid, RA also has some bioactive glycoside derivatives including rosmarinic acid-3-*O*-glucoside (salviaflaside), rosmarinic acid-3′-*O*-glucoside, rosmarinic acid-4-*O*-glucoside, rosmarinic acid-4′-*O*-glucoside, and rosmarinic acid-4,4′-*O*-diglucoside. For example, rosmarinic acid-4-*O*-glucoside has been studied with a pleiotropic effect against viral pneumonia: (1) To reduce the levels of inflammatory cytokine and oxidative stress in the serum and lungs of A/FM/1/47 H1N1 virus infected mice; and (2) to lower the tissue fluid into the alveoli and inhibit virus proliferation, improve ventilation, and reduce mortality [204].

Aside from these OH-substituted derivatives, there are a series of depside derivatives known as salvianolic acids. Salvianolic acid B is the most famous and representative one, which is listed as one of the chemical markers for the quality evaluation of *Salvia miltiorrhiza* Bge. roots and rhizomes in both the Chinese Pharmacopoeia and United States Pharmacopoeia. This phytochemical has revealed multiple bioactivities including (1) a protective effect on the brain from ischemia/reperfusion-induced injury by inhibiting reactive oxygen species (ROS)-mediated inflammation [205]; (2) a protective effect on the liver from acute and chronic injury by the inhibition of Smad2C/L phosphorylation [206]; (3) an anti-inflammatory effect on atherosclerosis through the mitogen-activated protein kinase/nuclear factor-κB (MAPKs/NF-κB) signaling pathways in vivo and in vitro [207]; (4) an anti-tumor effect against human breast cancer adenocarcinoma cells [208]; and (5) anti-diabetic effects [209]. 

All of these mentioned components are considered as the derivatives of biosynthesis from RA. Compared to conventional chemical extraction, inducers can be used to induce plant cells to synthesize valuable secondary metabolites, which is more economical and feasible and less likely to cause pollution. Therefore, it is urgent to explore the possible derivatization patterns and to elucidate the regulatory mechanism of secondary metabolism in these plants.

## 5. Analytical Technique

Characterized with the higher separation efficiency, less time and sample consumption, and a wider application range, high performance liquid chromatography (HPLC) or ultra performance liquid chromatography (UPLC) has gradually become the mainstream analytical technique in the research field of herbal plants, a complicated matrix with a variety of natural products. Due to its great conjugation system, RA has a strong absorbance in the ultraviolet region. Therefore, for the majority of research papers on the quantitative analysis of RA, an ultraviolet detector or diode-array detector was the mostly used [194,210,211,212]. Moreover, HPLC coupled with evaporative light scattering detector (ELSD) has also been applied for the quantitation of RA in *Rosmarinus officinalis* L. leaves [213]. However, in the biological samples, there are many endogenous interfering substances present and the content of the analyte is much lower. As a result, in the pharmacokinetic studies of RA concerning plasma, serum, or different tissues, a mass spectrometry detector with multiple-reaction monitoring mode has often been utilized [214,215,216,217].

Capillary electrophoresis (CE) is another widely-used and effective separation technique for the analysis of natural products. Many subtypes are inclusive in CE. However, capillary zone electrophoresis (CZE) and micellar electrokinetic chromatography (MEKC) are the main two used for RA quantitation. In the CZE experiments, a sodium borate solution was used as the run buffer to determine the RA in *Salvia officinalis* tea samples [218], in 14 Salvia species [219], in *Origanum Vulgare* L. [220], and in *Melissa officinalis* products [221]. In the MEKC studies, to obtain a satisfactory separation of RA from the other components, some additives were supplemented to the buffer such as β-cyclodextrin [222,223] and sodium dodecylsulfate [224].

However, it is well-known that some physicochemical pretreatments are necessary when the aforementioned LC or CE method is employed. In recent years, nondestructive determination methods have caused wide concern, of which some techniques related to infrared are the ones most representative. There have been some successful examples of the quantitative analysis of RA in *Rosmarinus officinalis* L. leaves [225], *Thymus vulgaris* L. or *Thymus zygis* L. leaves and flowers [226], and several Lamiaceae plants [227]. In these studies, the conventional HPLC method has also been used to compare the results along with partial least squares regression analysis, a chemometric model for calibration and validation.

## 6. Pharmacology

RA, a natural product from many plants, has been studied to possess a wide range of similar pharmacological activities with its origins such as anti-inflammation, anti-oxidation, anti-diabetes, anti-tumor, anti-virus, neuroprotection, hepatoprotection, and others in many in vivo and in vitro studies.

### 6.1. Anti-Inflammation

Inflammatory diseases are the pathological processes of defense responses evoked by some stimulation such as infection and trauma and are characterized by the imbalance in inflammatory mediators and cells. Inflammation also has a significant impact on human health and is involved in many other diseases. In recent decades, phytochemicals have attracted more and more attention regarding treatment.

In osteoarthritis, the degradation of cartilage extracellular matrix (ECM) might be induced by the depletion of collagen 2 and aggrecan, two of its main components. In addition, a disintegrin and metalloproteinase with thrombospondin motifs-4 (ADAMTS-4) and ADAMTS-5 are involved in this degradation. In an in vitro study of IL-1β-induced chondrocytes, the gene expression of collagen 2 and aggrecan were inhibited and ECM degradation occurred. RA incubation of 100 μM was observed to abolish this inhibition and demonstrate the inhibitory effect on IL-6 production, the gene and protein expression of ADAMTS-4 and ADAMTS-5, and even on the ECM degradation. The outcome led to the conclusion that RA may be a promising drug for osteoarthritis treatment [228]. In another in vivo study of the mice arthritis model induced by collagen, intraperitoneal injection of RA (50 mg/kg) markedly improved the arthritis index and reduced the affected paw number. Compared to those in the control group, severe leukocyte infiltration, the architecture of synovial tissues, and bone integrity loss were also more normal in the RA treatment group, manifesting a lower histopathologic index [229].

It is common knowledge that T cells are involved in atopic dermatitis (AD) pathogenesis. In the acute stage, AD skin lesions are infiltrated by CD4^+^ T cells, which could secrete IL-4, IL-5, and IL-13. In the chronic stage, Th1 cells secrete interferon-γ (IFN-γ). Some researchers have reported that RA (5 μM) could significantly inhibit the production of IL-4 and IFN-γ through activated CD4^+^ T cells. In addition, the same researchers also found after 2,4-dinitrofluorobenzene challenge, the symptoms of AD-like skin lesions were found on the NC/Nga mice such as pruritus, eruptions, and ear swelling. In this pathological state, the serum IgE level was tested as abnormally high and the characteristic dermal infiltration of inflammatory cells including CD4^+^ T, CD8^+^ T, and mast cells into ear skin lesions was observed to be markedly increased. Intraperitoneal administration of RA (50 mg/kg) also exhibited remarkable ameliorating and inhibiting effects on the above pathological phenomenon [230].

Inflammatory bowel disease is a chronic and recurrent intestinal inflammation in which ulcerative colitis is a typical one. In mice with colitis induced by dextran sulfate sodium, the oral administration of RA (60 mg/kg) significantly reduced the severity of colitis as shown by the disease activity index scores, colonic damage, and colon length. Furthermore, RA treatment also led to the decrease in some of the proinflammatory cytokines including IL-6, IL-1β, and IL-22, and the protein levels of cyclooxygenase-2 (COX-2) and inducible nitric oxide synthase (iNOS) in the colons. These protective effects were proven to be related to the inhibition of NF-κB and signal transducer and activator of transcription 3 (STAT3) activation [231]. In another study, RA was believed to protect from ulcerative colitis by regulating macrophage polarization depending on heme oxygenase-1 [232].

Aside from the above-mentioned studies, RA has been studied in vitro or in vivo to exert protective or ameliorative properties on lipopolysaccharide (LPS)-induced mastitis [233], sodium taurocholate-stimulated acute pancreatitis [234], LPS-induced acute lung injury [235], LPS-induced neuroinflammation [236], plaque-induced gingivitis [237], concanavalin A-induced hepatic injury [238], ovalbumin-stimulated allergic rhinitis [239], etc.

### 6.2. Anti-Oxidation

Combined challenge of ovalbumin and hydrogen peroxide would lead to a superimposed asthma with oxidative lung damage symptoms in mice. In the BALF and lung tissues of the model group, inflammatory cells including eosinophils, neutrophils, and lymphocytes and cytokines IL-4, IL-5, IL-13, and IFN-γ were all found to be elevated; ROS, nicotinamide adenine dinucleotide phosphate oxidase-2 (NOX-2), and NOX-4 levels were remarkably upregulated; and the total superoxide dismutase (SOD), total glutathione peroxidase (GSH-Px), catalase (CAT), and Cu/Zn SOD activities were observably downregulated compared to those in the blank group. RA treatment (10, 20, 40 mg/kg) exhibited alleviative and protective effects on the above symptoms and the highest dose was even more effective than dexamethasone [240]. In terms of antioxidant property in *Caenorhabditis elegans*, RA (60, 120, 180 μM) could significantly enhance the catalase activity, GSH-Px activity, and reduce glutathione (GSH) content and the glutathione/oxidized glutathione ratio as well as diminish the malondialdehyde (MDA) content in a dose-dependent manner, which resulted in promoting the lifespan and motoricity and reducing the fat store without threatening fertility. Furthermore, after RA treatment, the survival rate under acute oxidative and thermal stress was increased while intestinal lipofuscin accumulation was suppressed. This strong antioxidant activity was deemed to be related to regulating the insulin/insulin-like growth factor signaling (IIS) and MAPK pathways and activating the downstream antioxidant enzyme gene expression in *Caenorhabditis elegans* [241]. Chromium is known to cause severe toxicity in the liver and kidney tissue. In a potassium dichromate challenged rat model, RA (25 mg/kg) oral gavage of 60 days was observed to show s protective effect and reduce the oxidative damage in the two tissues. Oxidative stress evaluation demonstrated a remarkable increase in the GSH level and a notable decrease in the MDA level in the RA treatment group compared to those in the model group. Immunohistochemical studies and Rt-PCR analysis have confirmed that the result might be obtained via the Nrf2 pathway [242]. By activating the same nuclear factor erythroid-2 related factor 2 (Nrf2) pathway and increasing the downstream antioxidant enzyme activity, the oral administration of RA at 2 mg/kg could protect mouse intestines against high-fat diet-stimulated oxidative stress by preventing intestinal epithelial cell apoptosis [243].

### 6.3. Anti-Diabetes

Some in vitro studies have exhibited the anti-diabetic activity of RA. The polyphenolic acid was shown to have an inhibitory effect on *α*-glucosidase with an IC_50_ value of 33.0 μM, much lower than that of acarbose (131.2 μM), a marketed *α*-glucosidase inhibitory drug [123]. RA was also demonstrated to have a regulatory effect on glucose homeostasis. It was found that RA (5.0 µM) could activate adenosine 5′-monophosphate-activated protein kinase (AMPK) phosphorylation and increase the glucose uptake in L6 rat muscle cells, comparable to the maximum insulin (0.1 µM) and metformin (2.0 mM) [244].

In a dose-dependent manner, RA treatment (120–200 mg/kg, 7 days) showed a remarkable hypoglycemic effect in streptozotocin-induced type-1-like diabetic rats and significantly improved the glucose uptake and insulin sensitivity in high-fat diet-induced type-2-like diabetic rats. This beneficial effect against diabetes was believed to be related to both the decrease in phosphoenolpyruvate carboxykinase expression in the liver and the increase in glucose transporter-4 expression in the skeletal muscle [245]. In another study, the RA treatment (100 mg/kg, 30 days) of diabetic rats was found to have the effect of restoring the blood glucose level and regulating the levels of adiponectin and leptin. In addition, the diabetic pathology in hepatic parenchymal structures was also attenuated by the introduction of RA through histological and ultrastructural observations [246]. Na^+^/glucose cotransporter-1 (SGLT1) is considered as an important glucose transporter from intestinal lumen to blood. RA administration (14 days) could reverse the streptozotocin-induced SGLT1 protein increase and stabilize the fasting blood glucose level in diabetic rats [247].

### 6.4. Anti-Tumor

Breast cancer stem-like cells play an important role in the initiation, maintenance, and metastasis of breast cancer. RA coincubation (270 μM, 810 μM) could decrease their viability, inhibit their migration, and induce their apoptosis. RT-PCR analysis and immunoblot analysis showed that the two concentrations of RA treatment notably lowered the levels of mRNA and the protein of phosphorylation of smoothened and Glioma-associated oncogene homolog 1. Furthermore, RA treatment also led to the downregulation of B-cell lymphoma-2 (Bcl-2) expression and the upregulation of Bax expression. Therefore, the anti-tumor effect of RA might be related to the Bcl-2 and hedgehog signaling pathways [248]. Cao et al. employed intragastric administration of RA (150, 300 mg/kg) for 10 days to treat H22 tumor-bearing mice. It was unveiled that RA could effectively inhibit the tumor growth by regulating the secretion of inflammation and angiogenesis cytokines (IL-1β, IL-6, tumor necrosis factor-α (TNF-α), vascular endothelial growth factor, and transforming growth factor-β) and suppressing NF-κB p65 expression in the microenvironment [249].

Regarding the 1,2-dimethylhydrazine (DMH)-induced colon carcinogenesis in rats, many pathological phenomena have been easily found and tested in the liver and colon such as a large number of colonic tumors, decreased lipid peroxidation, antioxidant status and glutathione-S-transferase activity, md elevated cytochrome P450 content and p-nitrophenol hydroxylase activity, which were significantly reversed by RA (5 mg/kg). The pronounced effects indicated the possibility of RA as a chemopreventive agent against colon cancer [250]. In another DMH-stimulated rat model with colon carcinogenesis, oral supplementation with RA (5 mg/kg) also demonstrated a pronounced anti-tumor activity, probably due to the reduction in aberrant crypt foci formation and multiplicity, the suppression of fecal and colonic mucosal bacterial enzyme activities, and the improvement in circulatory thiobarbituric acid reactive substances (TBARS), enzymic, and non-enzymic antioxidant status [251].

In addition, RA also showed an anti-tumor effect on 7,12-dimethylbenz(a)anthracene-induced skin carcinogenesis [252], a cytotoxic effect on ARH-77 human (multiple myeloma) cells [253], prostate cancer cells [254], and human Hep-G2 liver carcinoma cells [255], and an inhibitory effect on the metastatic properties of colorectal cancer cells [256].

### 6.5. Anti-Virus

Enterovirus 71 (EV71) is a nonenveloped single-stranded RNA virus and easily causes hand, foot, and mouth disease, and even neurological complications or fatality in children. However, there is no specific and pointed treatment. Recently, phytomedicines and phytochemicals have been the alternative to chemical drugs for anti-virus. The *Melissa officinalis* extract was investigated to possess anti-EV71 activity and RA was identified and proven to be the responsible bioactive component therein, in which the alleviations of p38 kinase and epidermal growth factor receptor substrate 15 hyperphosphorylation were deduced to be involved [257]. In EV71-infected human rhabdomyosarcoma cells, RA was tested with a low IC_50_ value (4.33 μM) and a high therapeutic index (340) when the infection multiplicity was 1. Further investigation showed that RA could protect the cells from the cytopathic effects and apoptosis at the early stage of viral infection. In EV71-challenged neonatal mice, RA (20 mg/kg) also manifested the similar protective effect at the early stage, prolonging survival time and reducing mortality [258]. The results of another study were consistent with these findings and revealed the possible mechanism associated with virus-P-selectin glycoprotein ligand-1 and virus-heparan sulfate interactions [259]. The above findings indicated RA as a potential EV71 inhibitor in the initial stages of viral infection.

Japanese encephalitis virus (JEV) is a crucial cause of acute encephalopathy in children, targeting the central nervous system. With intraperitoneal treatment of RA (25 mg/kg), the significant reduction in the mortality of JEV-infected mice was observed, along with the dramatic decreases in viral loads and proinflammatory cytokines including IL-12, TNF-α, IFN-γ, monocyte chemotactic protein 1 (MCP-1), and IL-6. These findings suggest the potential of RA as a candidate for JEV treatment [260]. In primary human hepatocytes infected by the hepatitis B virus (HBV), RA exhibited an inhibitory effect on HBV replication and a potentiation effect on the anti-HBV activity of lamivudine [261]. Furthermore, like oseltamivir, RA (IC_50_ = 0.40 μM) showed high neuraminidase (NA)-inhibiting activity from an in vitro study of the anti-influenza virus, which was confirmed by the high binding affinity, hot-spot residues, and II-bond formations of the RA/NA complex from the in silico study [262].

### 6.6. Neuroprotection

6-Hydroxydopamine (6-OHDA) is known to be a neurotoxin used to create similar symptoms as Parkinson’s disease (PD). In MES23.5 dopaminergic cells co-incubated with 6-OHDA, RA (0.1 mM) could protect them from induced neurotoxicity through preventing the viability reduction and upregulating the ROS generation and mitochondria membrane potential [263]. In an in vivo study on the 6-OHDA-induced rat model, RA (20 mg/kg) through intragastric administration showed a neuroreparative function on the degeneration of the nigrostriatal dopaminergic system by decreasing the nigral iron level and regulating the Bcl-2/Bax gene expression [264]. Therefore, regarding PD, RA might be viewed as a therapeutic treatment for related patients in the future.

Aβ42 was used to induce an Alzheimer’s disease-like rat model, resulting in a significant increase in the levels of TBARS and 4-hydroxy-2-nonenal and decrease in the SOD, CAT, GSH-Px, and glutathione levels with the reduction in acetylcholine content and acetylcholine esterase activity. In addition, mismatch negativity response and θ power/coherence of auditory event related potentials were also decreased. Fortunately, RA (50 mg/kg, oral administration) demonstrated an attenuating effect on these observed pathological changes and the increased Aβ staining and astrocyte activation [265]. In a kainate-induced rat model, seizure intensity, apoptosis, oxidative stress markers (MDA, GSH, CAT), Timm index, and the number of Nissl-stained neurons were employed as the indicators to evaluate the beneficial effect of RA. The results supported the neuroprotective effect of RA (10 mg/kg) against temporal lobe epilepsy [266]. Intraperitoneal administration of RA (20 mg/kg) was observed to improve the working, spatial, and recognition memory deficits, and to reduce the infarct size and neurological deficits of the rats lesioned by permanent middle cerebral artery occlusion, which were speculated to be related to suppressing neuronal loss and increasing synaptophysin expression and brain-derived neurotrophic factor. Therefore, the results indicate the memory protective effect of RA [267]. As for spinal cord injury, RA was investigated to show a neuroprotective effect on this severe central nervous system injury through inhibiting the TLR4/NF-κB pathway and activating the Nrf2/HO-1 pathway, as witnessed by in vitro (55.6 μM) and in vivo (40 mg/kg, intraperitoneal administration) studies [268].

### 6.7. Hepatoprotection

Li et al. conducted in vitro and in vivo studies to observe the hepatoprotective effect of RA against experimental liver fibrosis. In hepatic stellate cells, RA co-incubation (32 μM) was found to inhibit cell proliferation and the expressions of transforming growth factor-β1 (TGF-β1), connective tissue growth factor (CTGF), and α-smooth muscle actin. In CCl_4_-intoxicated rats with liver fibrosis, RA (10 mg/kg) could reduce the fibrosis grade, ameliorate biochemical indicators (albumin, globulin, alanine aminotransferase, glutamate-pyruvate transaminase) and histopathological morphology, and downregulate the liver TGF-β1 and CTGF expression [269]. The findings were then witnessed and confirmed by Domitrovic and his colleagues in a mice model with CCl_4_-intoxicated liver fibrosis. In addition to improvements in the histological and serum markers concerning liver damage and the inhibition of TGF-β1 and CTGF expression, the amelioration of oxidative/nitrosative stress and inflammatory response (NF-κB, TNF-α, COX-2) and the upregulation of Nrf2 and heme oxygenase-1 expression were also found after RA treatment (50 mg/kg) [270].

In the bile duct ligation-induced extrahepatic cholestasis rat model, RA (20 mg/kg) exhibited a hepatoprotective effect by alleviating TGF-β1 production and hepatic collagen deposition and ameliorating hepatic inflammation. Resolution of oxidative burden and downregulation of high mobility group box-1/toll-like receptor-4 (HMGB1/TLR4), NF-κB, AP-1, and TGF-β1/Smad signaling were investigated to be involved in RA hepatoprotection [271]. Furthermore, Lou et al. used the partial hepatectomy model to explore the effects of RA on liver regeneration. The evaluation content included the index of the liver to body weight and the expression of proliferating cell nuclear antigen and liver transaminases. As a result, RA (200 mg/kg) could promote liver regeneration and restore lesioned liver function via the mammalian target of rapamycin/S6 protein kinase (mTOR/S6K) pathway [272]. Furthermore, in a mouse model of non-alcohol steatohepatitis induced by a methionine-choline-deficient diet, RA (10 mg/kg) exhibited a remarkable hepatoprotective potential by decreasing the plasma triglyceride, cholesterol, liver steatosis, and oxidative stress, which was deemed to be related to the activation of the silent information regulator-two 1 (SIRT1)/Nrf2, SIRT1/NF-κB, and SIRT1/peroxisome proliferator-activated receptor α (PPARα) pathways [273].

### 6.8. Other Activities

To study RA protection against premature ovarian failure (POF), the intraperitoneal injection of cyclophosphamide was used to induce the mouse model. With the help of fluorescence immunohistochemistry, histological analysis, Western blot analysis and polymerase chain reaction, RA (40 mg/kg) was investigated to effectively attenuate the abnormal situations of the model including injured ovarian, increased ovarian index, and serum sex hormone levels, the overexpression of the nucleotide-binding oligomerization domain receptor protein-3 (NLRP3) inflammasome, and apoptosis-related proteins in the ovarian. The findings indicate that RA might have a bright prospect in POF treatment in the future [274].

In the treatment of thoracic tumor, radiotherapy is an essential therapy method, which will unfortunately cause pulmonary fibrosis later. Zhang et al. observed that RA (120 mg/kg) could regulate NF-κB signaling and the RhoA/Rock pathway through microRNA-19b-3p, which were responsible for the alleviation of inflammatory reactions, the reduction in collagen hyperplasia, and the suppression of pulmonary fibrosis development in the X-ray irradiation-induced rat model. Thus, RA is believed to be a potential alternative to attenuating radiotherapy-caused pulmonary fibrosis [275].

Ji et al. established aa high-fat diet and VD_3_-induced rat model to observe the effect of RA on vascular calcification. The results showed that RA (200 mg/kg) could notably decrease the levels of alkaline phosphatase, phosphorus, calcium, MDA, increase the SOD level, and reduce the calcified nodule content and ROS production. Additionally, the levels of Nrf2, heme oxygenase-1, NAD(P)H quinone dehydrogenase, and osteoprotegerin were upregulated, while the levels of kelch-like ECH-associated protein 1, NF-κB, β-catenin, and osteogenic transcription factor were significantly downregulated. RA coincubation (80 μM) also showed similar effects in the β-glyerophosphate-induced rat aortic smooth muscle cell model. These functions in improvement were proven to be related to the regulation of the Nrf2 pathway [276].

## 7. Clinical Studies

Although RA has been proven to have potential for drug application in many research articles, only two papers have been published on the clinical study of RA as a pure compound.

It was reported that there were 14 female and seven male patients with mild AD inclusive in a clinical study, in which a RA (0.3%) emulsion was applied to the elbow flexures twice a day. Compared to before the treatment, erythema and transepidermal water loss of the antecubital fossa were reduced notably after treatment of four or eight weeks. Self-reports from the patients showed that dryness, pruritus, and general AD symptoms were ameliorated after RA smearing [277]. Another clinical study enrolled 29 patients with seasonal allergic rhinoconjunctivitis. The results indicated that RA oral treatment (80, 200 mg/kg) led to significant decreases in the incidence rates for itchy nose, watery eyes, itchy eyes, and total symptoms compared to the placebo. Meanwhile, the number of neutrophils and eosinophils in the nasal lavage fluid were also significantly decreased [278].

## 8. Applications in Food Science

It is well-known that the polyphenol natural products are characterized by their multiple phenolic hydroxyl groups and accompanying anti-oxidation. As an organic acid with four phenolic hydroxyl groups, RA has exhibited its anti-oxidant capacity not only in pharmacological studies, but also in food scientific studies.

In sea buckthorn fruit wine, concerning DPPH radical scavenging and hydroxide radical scavenging, RA showed a greater antioxidant capacity (IC_50_ = 8.02 mg/mL, 99.31% clearance rate) than sulfur dioxide (IC_50_ = 10.31 mg/mL, 98.67% clearance rate), the conventional antioxidant. Therefore, RA was speculated to be an ideal antioxidant alternative to sulfur dioxide in wine fermentation due to its safety and stability [279]. Li et al. prepared different rabbit skin gelatin–RA composite films to study their preservation effects on the pork quality during cold storage. As a result, a composite film with RA of 0.8 g/L could effectively inhibit the increase in the total number of colonies, total volatile basic nitrogen content and pH, extend the shelf life of pork from 4 to 8 days, remain at a high hardness, and reasonable chromatic aberration. Therefore, the rabbit skin gelatin–RA composite film was proposed to be a potential packaging material for the preservation and freshness of pork [280]. A total of 1% chitosan containing 30 mg/L RA was studied with total viable counts (less than 6.0 log CFU/g), potassium value (less than 60%), free fatty acids (2.5%), trimethylamine (2 mg/100 g), and H_2_S-producing bacteria (less than 6.0 log CFU/g) and to maintain better sensory characteristics and flavor quality of the half-smooth tongue sole fillets stored at 4 °C for 18 days. On the other hand, the results for the control group were 7.5 log CFU/g, nearly 90%, 5.0%, and 3.7 mg/100 g more than 6.0 log CFU/g, respectively. Therefore, these significant differences indicate the complex potential to improve the quality of this fish during refrigerated storage [281].

Therefore, as a polyphenolic acid with antioxidant and antibacterial activity, RA could retard the growth of microorganisms and inhibit the increase in the pH value and perioxidation in food, contributing to its capacity of keeping food quality, slowing decay, and extending the shelf life.

## 9. Pharmacokinetics

It is well-known that the pharmacokinetic profiles are fundamental for a potential candidate drug. Regarding RA with the pronounced bioactivities, it is exactly that. A study of this polyphenolic acid was carried out on the metabolites and the pharmacokinetic pathways in normal rats. As a result, a total of 36 metabolites including RA itself were identified in plasma, urine, and feces after oral administration. The prototype and glucuronic acid conjugation were found to be predominant in plasma. Furthermore, Phase I metabolism (primarily hydrolysis) and Phase II metabolism (sulfation, methylation, glucuronic acid conjugation, and glucose conjugation) were mainly involved in the feces and urine, respectively [282]. In another study associated with human liver microsomes, after 1 h of incubation, RA was transformed to yield 14 metabolites and several metabolic pathways were speculated including oxidation, glucuronic acid conjugation, hydroxylation, and GSH conjugation [283].

To reveal the oral absolute bioavailability of RA, the normal rats were administered with the phytochemical through intragastrical (12.5, 25, 50 mg/kg) and intravenous (0.625 mg/kg) methods. The calculated parameters showed rapid absorption and middle-speed elimination for the pharmacokinetic characters of RA after oral administration in rats. In addition, poor absolute bioavailability was demonstrated with 1.69%, 1.28%, and 0.91% for 12.5 mg/kg, 25 mg/kg, and 50 mg/kg, respectively [284]. In a hepatoprotective and metabolic study, RA treatment could significantly suppress the pathological changes in the bile rate, thiobarbituric acid (TBA), total bilirubin (TBIL), alanine aminotransferase (ALT), and aspartate aminotransferase (AST) of rats with cholestatic liver injury. On the other hand, cholestasis resulted in PK behavior variations and the drug accumulation of RA, which were witnessed by the decrease of 14.5% for CL and the increase of 17.0% for AUC_(0__→__∞)_, 40.3% for T_max_, and 13.1% for C_max_, compared to those of normal rats [285]. As above-mentioned, RA often serves as an indicator for the quality evaluation of some folk herbal medicines or compound medicines. In the same way, RA also acts as a representative component in the pharmacokinetic studies of these medicines such as the *Salvia miltiorrhiza* polyphenolic acid solution [286], *Prunella vulgaris* extract [287], ZibuPiyin Recipe [288], and Xuebijing Injection [289]. Table 2 presents the specific parameters of these pharmacokinetic studies including the AUC_(0__→__∞)_, T_max_, C_max_, and CL.

As a potential candidate drug with various effects, RA should first be based on its toxicity. However, there have only been several in vitro studies mentioning its non-cytotoxicity at the test concentrations in normal cells such as chondrocytes (100 μM) [228], HepG2 cells (100 μM) [290], N2A mouse neuroblastoma cells (250 μM) [291], and A172 human astrocytes (83 μM) [292], with the median lethal concentration in zebrafish embryos of 296.0 μM [236]. In the two clinical studies, it was reported that there were no self-feeling adverse events and no significant abnormalities in routine blood tests [277,278]. Therefore, a systematic toxicity investigation of RA should be conducted urgently in the future including acute toxicity, chronic toxicity, LD_50_, therapeutic window, etc. Additionally, according to the administration approach of the phytochemical, different types of test animals should be involved, where each important tissue and organ should be observed, and each blood index should be tested. After all, safety is the first key character of a drug, especially prior to its clinical application.

## 10. Future Perspectives

As above-mentioned, RA is believed to be a polyphenolic acid that widely occurs in natural plants, especially from Lamiaceae. During over the past sixty years, RA has exhibited miscellaneous pharmacological activities, pharmacokinetic characteristics, and a variety of natural sources and derivatives.

Generally speaking, the structural modification of a natural product often aims to improve its bioavailability, to extend (improve) its bioactivities, or to diminish its toxicity. Up to now, many RA derivatives have been found in nature and some have revealed impressive biological effects, which could be considered as the products of RA structural modification. However, it was unordered and unscheduled in the way in which they were isolated and found biological. With the further understanding of the RA action mechanism, structural modification with specific purposes should be well-designed and carried out in the future including (1) investigating the RA chemical structure by crystallology and quantum mechanics; (2) simulating the combination of the RA and target from the protein database; (3) summarizing the action rules of the RA derivatives with different substituent groups; (4) systematically proving their bioactivities with high throughput screening.

In terms of pharmacokinetic study of RA, there have been many articles reported including the intragastrical administration of this pure component or some compound formulations, its application on the test animals or humans, and its application on normal or model animals. However, there still exist some issues worth discussing. (1) The rats, especially the normal ones, were used in the majority of pharmacokinetic studies. Now that RA has showed a variety of pharmacological activities, the corresponding model animals should be the first choices. Additionally, rats should not be the only test species. (2) A single dose of RA administration was involved in a large number of pharmacokinetic studies. Since RA will act as a candidate drug and the treatment will last for several days, it seems that a multi-dose of RA administration is necessary and the relevant pharmacokinetic studies are essential. (3) The number of clinical pharmacokinetic studies of RA is small [293,294]. At present, RA is not approved as a legal drug and is prohibited for medical application on humans. However, in traditional medicines, some herbal extracts or compound formulations enriched with RA are allowable. Their pharmacokinetic studies could provide some basis for further drawing of the RA pharmacokinetic profile. (4) Intragastrical administration was the main focus while other administration methods have been rarely investigated and would be interesting to pursue in the future.

Until now, RA has been well-acknowledged as a promising natural product with a variety of pharmacological activities such as anti-oxidation, anti-inflammation, anti-tumor, anti-virus, anti-diabetes, etc. Furthermore, some possible signaling pathways have been explored. However, to be developed as a true candidate drug, RA should be investigated with a focus on some straightforward and effective bioactivity for some diseases including the drug-delivery method, therapeutic dose, and possible action mechanism. On the other hand, based on the clear action targets and the results of the in vitro studies, computer-assisted molecular docking is becoming a virtual screening method for both drugs and their bioactivities. RA has been found to inhibit peptide deformylase, N-myristoyltransferase, human hyaluronidase enzyme, and influenza neuraminidase through in silico evaluations [262,295]. Therefore, in the future, this technology would help us discover more activities and widen the medical application range of RA.

RA is a polyphenolic acid characterized by poor lipid solubility, poor membrane permeability, and low oral absolute bioavailability, which has limited its application. Some liposomes and solid lipid nanoparticles have been revealed to be promising [296,297]. Therefore, aside from structural modification, some systematic studies concerning pharmaceutical formulations or special excipients should be carried out to avoid RA degradation in the gastrointestinal tract and to transport RA to the target tissues. With these achievements, the shortcomings of limited absorption, fast distribution, fast metabolism, and fast elimination might be overcome in the future. Meanwhile, the present research of RA in food science are around anti-oxidation and maintaining the food color and luster. However, it is important to explore the possibility of RA being used as an alternative to the traditional additives. Therefore, the study hotspots should be to compare this phytochemical and the main additives not only affecting the food quality, but also in its safe use.

## 11. Conclusions

Taken together, all the research findings indicate that RA is a candidate drug or a lead component naturally occurring in plants. In the present paper, we summarized the achievements from phytochemistry, pharmacology, pharmacokinetics, and other study aspects of RA and proposed some interesting issues worth investigating in the future. We hope this paper can help researchers either in fundamental research or in applied research to understand RA more comprehensively, utilize RA more efficiently, and eventually develop RA as a novel drug.

## Figures and Tables

**Figure 1 molecules-27-03292-f001:**
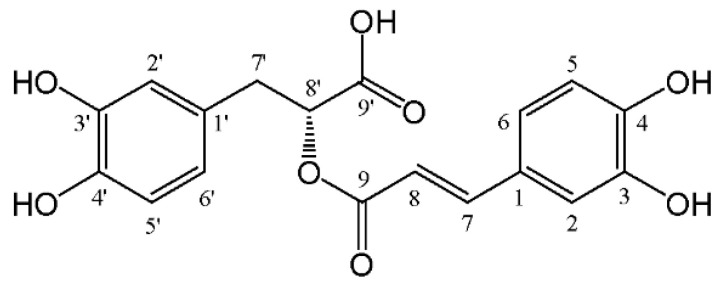
The chemical structure of RA.

**Figure 2 molecules-27-03292-f002:**
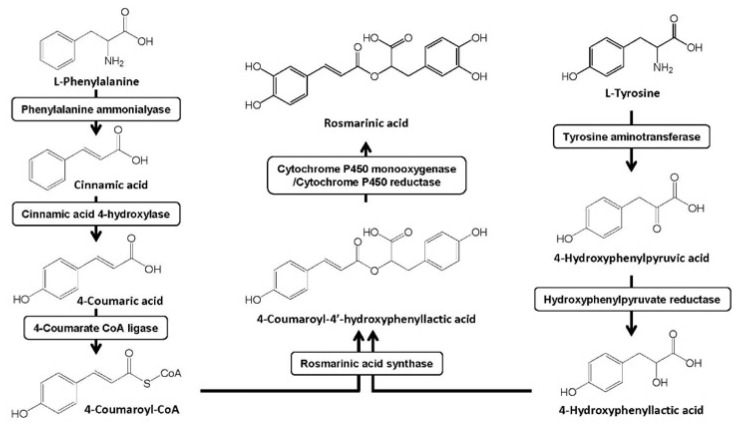
Biosynthesis pathway of RA in the plants.

**Table 1 molecules-27-03292-t001:** Plants containing RA.

No.	Plant	Family	Part	Reference
1	*Adenium obesum*	Apocynaceae	Stem Barks	[11]
2	*Alkanna sfikasiana* Tan, Vold and Strid	Boraginaceae	Roots	[12]
3	*Anchusa azurea* Miller var. azurea	Boraginaceae	Roots	[13]
4	*Anchusa italica* Retz.	Boraginaceae	-	[14]
5	*Anchusa strigosa* Banks et Sol	Boraginaceae	Roots	[15]
6	*Anthoceros punctatus*	Anthocerotaceae	-	[16]
7	*Apeiba tibourbou* Aubl.	Tiliaceae	Leaves	[17]
8	*Arctopus monacanthus*	Apiaceae	Roots	[18]
9	*Arnebia purpurea* S. Erik & H. Sumbul	Boraginaceae	Roots	[19]
10	*Baccharis chilco* Kunth	Asteraceae	Aerial parts	[20]
11	*Barbarea integrifolia*	Brassicaceae	Aerial parts	[21]
12	*Bellis sylvestris*	Asteraceae	Leaves	[22]
13	*Blechnum brasiliense*	Blechnaceae	Leaves	[23]
14	*Canna edulis* Ker	Cannceae	Rhizomes	[24]
15	*Celastrus hindsii* Benth	Celastraceae	Leaves	[25]
16	*Centella asiatica*	Apiaceae	Aerial parts	[26]
17	*Chloranthus fortune* (A. Gray) Solms-Laub	Chloranthaceae	Whole plants	[27]
18	*Chloranthus multistachys* Pei	Chloranthaceae	-	[28]
19	*Clerodendranthus spicatus* (Thunb.) C.Y. Wu	Lamiaceae	Whole plants	[29]
			Aerial parts	[30]
20	*Clinopodium chinense* var. parviflorum	Lamiaceae	Aerial parts	[31]
21	*Clinopodium tomentosum* (Kunth) Govaerts	Lamiaceae	Aerial parts	[32]
22	*Clinopodium urticifolium*	Lamiaceae	Whole plants	[33]
23	*Coleus aromaticus* Benth.	Lamiaceae	Leaves	[34]
24	*Coleus forskohlii* (Willd) Briq.	Lamiaceae	Whole plants	[35]
25	*Coleus parvifolius* Benth.	Lamiaceae	Aerial parts	[36]
26	*Colocasia esculenta* (L.) Schott	Araceae	Leaves	[37]
27	*Cordia alliodora*	Boraginaceae	Root barks	[38]
28	*Cordia bicolor*	Boraginaceae	Leaves	[39]
29	*Cordia boissieri* A. DC.	Boraginaceae	Leaves	[40]
30	*Cordia dentata*	Boraginaceae	Leaves	[39]
31	*Cordia latifolia* Roxb.	Boraginaceae	Fruits	[41]
32	*Cordia megalantha*	Boraginaceae	Leaves	[39]
33	*Cordia morelosana* Standley	Boraginaceae	Aerial parts	[42]
34	*Cordia sinensis*	Boraginaceae	Whole plants	[43]
35	*Cordia verbenacea*	Boraginaceae	Leaves	[44]
36	*Cynoglossum columnae* Ten.	Boraginaceae	Roots	[45]
37	*Dracocephalum fruticulosum* Steph. Ex Willd.	Lamiaceae	Aerial parts	[46]
38	*Dracocephalum heterophyllum*	Lamiaceae	Whole plants	[47]
39	*Dracocephalum nutans* L.	Lamiaceae	Aerial parts	[46]
40	*Dracocephalum palmatum* Stephan	Lamiaceae	Aerial parts	[48]
41	*Dracocephalum tanguticum* Maxim.	Lamiaceae	Whole plants	[49]
42	*Ehretia asperula*	Boraginaceae	Leaves	[50]
43	*Ehretia obtusifolia*	Boraginaceae	Whole plants	[51]
44	*Ehretia philippinensis*	Boraginaceae	Barks	[52]
45	*Ehretia thyrsiflora*	Boraginaceae	Leaves	[53]
46	*Elsholtiza bodinieri* Vaniot	Lamiaceae	Whole plants	[54]
47	*Elsholtzia rugulosa* Hemsl.	Lamiaceae	Aerial parts	[55]
48	*Elsholtzia splendens* Nakai	Lamiaceae	Flowers and leaves	[56]
49	*Farfugium japonicum* (L.) Kitam. Var. *giganteum* (Siebold et Zucc.) Kitam.	Asteraceae	Flowers	[57]
50	*Foeniculum vulgare* Mill.	Apiaceae	Aerial parts	[58]
51	*Forsythia koreana* Nakai	Oleaceae	Fruits	[59]
52	*Gastrocotyle hispida*	Boraginaceae	Aerial parts	[60]
53	*Glechoma longituba*	Lamiaceae	Whole plants	[61]
54	*Hamelia patens* Jacq.	Rubiaceae	Aerial parts	[62]
55	*Hedera helix* L.	Araliaceae	-	[63]
56	*Helicteres angustifolia* Linn.	Sterculiaceae	Roots	[64]
57	*Helicteres hirsuta* Lour	Sterculiaceae	Stems	[65]
58	*Helicteres isora* L.	Sterculiaceae	Fruits	[66]
59	*Hypenia salzmannii* (Benth.) Harley	Lamiaceae	Leaves	[67]
60	*Hyptis atrorubens* Poit.	Lamiaceae	Leaves and stems	[68]
61	*Hyptis capitata* Jacq.	Lamiaceae	Aerial parts	[69]
62	*Hyptis pectinata* (L.) Poit	Lamiaceae	Leaves	[70]
63	*Hyptis suaveolens* (L.) Poit	Lamiaceae	Aerial parts	[71]
64	*Hyptis verticillata* Jacq.	Lamiaceae	Aerial parts	[72]
65	*Hyssopus cuspidatus*	Lamiaceae	Whole plants	[73]
66	*Ipomoea turpethum* (L.) R.Br.	Convolvulaceae	Whole plants	[74]
67	*Isodon eriocalyx* (Dunn) Hara var. *laxiflora* C. Y. Wu et H. W. Li	Lamiaceae	Leaves	[75]
68	*Isodon flexicaulis* C. Y. Wu et H. W. Li	Lamiaceae	Aerial parts	[76]
69	*Isodon lophanthoides* var. *graciliflorus*	Lamiaceae	Aerial parts	[77]
			Leaves	[78]
*70*	*Isodon oresbius* (W. W. Smith) Kudo	Lamiaceae	Aerial parts	[79]
71	*Isodon rubescens* (Hemsl.) Hara	Lamiaceae	-	[80]
72	*Isodon rugosus* (Wall. Ex Benth.) Codd	Lamiaceae	Aerial parts	[81]
73	*Isodon sculponeata* (Vaniot) Hara.	Lamiaceae	Leaves	[82]
74	*Keiskea japonica* Miq.	Lamiaceae	Aerial parts	[83]
75	*Lallemantia iberica* (Bieb.) Fisch & C.A. Mey	Lamiaceae	Aerial parts	[84]
76	*Lavandula angustifolia* Mill.	Lamiaceae	Aerial parts	[85]
77	*Lepechinia graveolens* (Reg.) Epling.	Lamiaceae	-	[86]
78	*Lepechinia meyenii* (Walp.) Epling	Lamiaceae	-	[87]
79	*Lepechinia speciosa* (St. Hill) Epling	Lamiaceae	-	[88]
80	*Lycopus europaeus* L.	Lamiaceae	Whole plants	[89]
81	*Lycopus lucidus* Turcz.	Lamiaceae	Aerial parts	[90]
82	*Marrubium vulgare* L.	Lamiaceae	Leaves	[91]
83	*Meehania urticifolia* (Miq.) Makino	Lamiaceae	Whole plants	[92]
84	*Melissa officinalis* L.	Lamiaceae	Aerial parts	[93]
			Leaves	[94]
85	*Mentha dumetorum*	Lamiaceae	Aerial parts	[95]
86	*Mentha haplocalyx* Briq.	Lamiaceae	Aerial parts	[96]
87	*Mentha longifolia* (L.) Hudson subsp. *longifolia*	Lamiaceae	Aerial parts	[97]
88	*Mentha piperita* L.	Lamiaceae	Leaves	[98]
			Aerial parts	[99]
89	*Mentha spicata* L.	Lamiaceae	Whole plants	[100]
90	*Mesona chinensis* Benth.	Lamiaceae	Whole plants	[101]
91	*Micromeria myrtifolia* Boiss. & Hohen	Lamiaceae	Aerial parts	[102]
92	*Microsorum fortunei* (T. Moore) Ching	Polypodiaceae	Leaves and stems	[103]
93	*Momordica balsamina*	Cucurbitaceae	Aerial parts	[104]
94	*Nepeta asterotricha* Rech. F.	Lamiaceae	Aerial parts	[105]
95	*Nepeta cadmea* Boiss.	Lamiaceae	Aerial parts	[106]
96	*Nepeta curviflora* Boiss.	Lamiaceae	Aerial parts	[107]
97	*Ocimum campechianum* Mill.	Lamiaceae	Leaves	[108]
98	*Ocimum sanctum* Linn.	Lamiaceae	Leaves and stems	[109]
99	*Origanum dictamnus* L.	Lamiaceae	Aerial parts	[110]
100	*Origanum glandulosum* Desf	Lamiaceae	Aerial parts	[111]
101	*Origanum majorana* L.	Lamiaceae	Aerial parts	[112]
102	*Origanum minutiflorum*	Lamiaceae	Aerial parts	[113]
103	*Origanum rotundifolium* Boiss.	Lamiaceae	Aerial parts	[114]
104	*Origanum vulgare* L. ssp. Hirtum	Lamiaceae	Aerial parts	[115]
105	*Paris veriticillata* Bieb.	Liliaceae	Roots	[116]
106	*Perilla frutescens* (L.) Britton var. *acuta* Kudo	Lamiaceae	Leaves	[117]
			Seeds	[118]
107	*Perilla frutescens* var. acuta	Lamiaceae	Fruits	[119]
108	*Perovskia atriplicifolia* Benth.	Lamiaceae	Roots	[120]
109	*Plectranthus forsteri* ‘Marginatus’	Lamiaceae	Aerial parts	[121]
110	*Plectranthus hadiensis* var. tomentosus	Lamiaceae	Aerial parts	[122]
111	*Plectranthus madagascariensis* (Pers.) Benth	Lamiaceae	Aerial parts	[123]
112	*Plectranthus scutellarioides* (L.) R. Br.	Lamiaceae	Aerial parts	[124]
113	*Polygomun aviculane*	Polygonaceae	Aerial parts	[125]
114	*Prunella vulgaris* L.	Lamiaceae	Spikes	[126]
115	*Prunella vulgaris* var. *lilacina*	Lamiaceae	Spikes	[127]
			Aerial parts	[128]
116	*Quercus serrata* Murray	Fagaceae	Leaves	[129]
117	*Rosmarinus officinalis* L.	Lamiaceae	Sprigs	[130]
			Leaves	[131]
118	*Salvia absconditiflora* Greuter & Burdet	Lamiaceae	Aerial parts	[132]
119	*Salvia castanea* Diels f. tomentosa Stib.	Lamiaceae	Rhizomes	[133]
120	*Salvia cavaleriei* Levi.	Lamiaceae	Whole plants	[134]
121	*Salvia cerino-pruinosa* Rech. F. var. *cerino-pruinosa*	Lamiaceae	Aerial parts	[135]
122	*Salvia chinensis* Benth.	Lamiaceae	Aerial parts	[136]
			Whole plants	[137]
123	*Salvia deserta* Schang	Lamiaceae	Roots	[138]
			Flowers	[139]
124	*Salvia flava* Forrest	Lamiaceae	Whole plants	[140]
125	*Salvia grandifolia* W. W. Smith	Lamiaceae	Roots	[141]
126	*Salvia kiaometiensis* Lévl.	Lamiaceae	Roots	[142]
127	*Salvia limbata* C.A. Meyer	Lamiaceae	Aerial parts	[143]
128	*Salvia miltiorrhiza* Bunge	Lamiaceae	Leaves	[144]
			Roots	[145]
*129*	*Salvia officinalis*	Lamiaceae	-	[146]
130	*Salvia palaestina* Bentham	Lamiaceae	Aerial parts	[147]
131	*Salvia plebeia* R. Br.	Lamiaceae	Leaves	[148]
			Whole plants	[149]
132	*Salvia przewalskii* Maxim	Lamiaceae	Roots and rhizomes	[150]
			Roots	[151]
133	*Salvia sonchifolia* C.Y. Wu	Lamiaceae	Roots	[152]
134	*Salvia splendens* Sellow ex Roem & Schult	Lamiaceae	Leaves	[153]
135	*Salvia trichoclada* Bentham	Lamiaceae	Whole plants	[154]
136	*Salvia viridis* L. cvar. Blue Jeans	Lamiaceae	Aerial parts	[155]
137	*Salvia yunaansis*	Lamiaceae	Roots	[156]
*138*	*Sanicula europaea* L.	Apiaceae	Aerial parts	[157]
139	*Sanicula lamelligera* Hance	Apiaceae	Whole plants	[158]
140	*Sarcandra glabra* (Thunb.) Nakai.	Chloranthaceae	Whole plants	[159]
141	*Satureja biflora*	Lamiaceae	Aerial parts	[160]
142	*Schizonepeta tenuifolia* Briquet	Lamiaceae	Aerial parts	[161]
143	*Sideritis albiflora*	Lamiaceae	Aerial parts	[162]
144	*Sideritis leptoclada*	Lamiaceae	Aerial parts	[162]
145	*Solanum betaceum* Cav.	Solanaceae	Fruits	[163]
146	*Solenostemon monostachys* Briq	Lamiaceae	Aerial parts	[164]
147	*Symphytum officinale* L.	Boraginaceae	Roots	[165]
148	*Thunbergia laurifolia Lindl*	Acanthaceae	Leaves	[166]
149	*Thymus alternans* Klokov	Lamiaceae	Aerial parts	[167]
150	*Thymus atlanticus* (Ball) Roussine	Lamiaceae	Leaves	[168]
151	*Thymus praecox* subsp *grossheimii* (Ronniger) Jalas	Lamiaceae	Aerial parts	[169]
152	*Thymus praecox* subsp *grossheimii* (Ronniger) Jalas var. *grossheimii*	Lamiaceae	Aerial parts	[170]
153	*Thymus quinquecostatus* var. *japonica*	Lamiaceae	Aerial parts	[171]
154	*Thymus serpyllum*	Lamiaceae	Whole plants	[172]
155	*Thymus sibthorpii* Bentham	Lamiaceae	Aerial parts	[173]
156	*Thymus sipyleus* subsp. *Sipyleus* var. *sipyleus*	Lamiaceae	Aerial parts	[174]
157	*Thymus vulgaris* L.	Lamiaceae	Leaves	[175]
158	*Tournefortia sarmentosa* Lam.	Boraginaceae	Stems	[176]
159	*Veronica sibirica* L.Pennell	Scrophulariaceae	Rhizomes	[177]
160	*Ziziphora clinopodioides* Lam.	Lamiaceae	Aerial parts	[178]
161	*Zostera marina*	Potamogetonaceae	Leaves	[179]
162	*Zostera noltii*	Potamogetonaceae	Leaves	[180]

-: not mentioned.

**Table 2 molecules-27-03292-t002:** The pharmacokinetic characteristics of RA in different test drugs and animals.

No.	Drug	Animal	Administration Mode	Pharmacokinetic Characters	Reference
1	RA	Normal rats	Intragastrical administration, 12.5 mg/kg	AUC_(0__→__∞)_ = 866.51 ng/mL·h, T_max_ = 0.139 h, C_max_ = 215.21 ng/mL, CL = 15.00 L/(h·kg)	[284]
			Intragastrical administration, 25 mg/kg	AUC_(0__→__∞)_ = 1308.62 ng/mL·h, T_max_ = 0.181 h, C_max_ = 361.57 ng/mL, CL = 19.20 L/(h·kg)	
			Intragastrical administration, 50 mg/kg	AUC_(0__→__∞)_ = 1866.58 ng/mL·h, T_max_ = 0.306 h, C_max_ = 790.96 ng/mL, CL = 27.60 L/(h·kg)	
			Intravenous administration, 0.625 mg/kg	AUC_(0__→__∞)_ = 2556.14 ng/mL·h, C_max_ = 6166.89 ng/mL, CL = 6.00 L/(h·kg)	
2	RA	Cholestatic liver injured rats	Intragastrical administration, 100 mg/kg	AUC_(0__→__∞)_ = 23.984 mg/mL·h, T_max_ = 0.988 h, C_max_ = 2.876 mg/mL, CL = 4.169 L/(h·kg)	[285]
		Normal rats		AUC_(0__→__∞)_ = 20.500 mg/mL·h, T_max_ = 0.704 h, C_max_ = 2.542 mg/mL, CL = 4.876 L/(h·kg)	
3	*Salvia miltiorrhiza* polyphenolic acid solution	Normal rats	Pulmonary administration, 10 mg/kg	AUC_(0__→__∞)_ = 200.01 ng/mL·h, T_max_ = 0.07 h, C_max_ = 370.78 ng/mL, CL = 0.05 L/(h·kg)	[286]
			Intravenous administration, 10 mg/kg	AUC_(0__→__∞)_ = 209.34 ng/mL·h, T_max_ = 0.03 h, C_max_ = 1344.10 ng/mL, CL = 0.05 L/(h·kg)	
4	*Prunella vulgaris* extract	Normal rats	Intragastrical administration, 10 mL/kg (1.25 mg/mL for RA)	AUC_(0__→__∞)_ = 737.7 ng/mL·h, T_max_ = 1.5 h, C_max_ = 120.8 ng/mL, CL = 21.0 L/(h·kg)	[287]
5	ZibuPiyin Recipe	Normal rats	Intragastrical administration, 3.951 g/kg (0.03 mg/g for RA)	AUC_(0__→__∞)_ = 3099.4 μg/mL·h, T_max_ = 1.7 h, C_max_ = 222.7 ng/mL	[288]
6	Xuebijing Injection	Normal rats	Intravenous administration, 6 mL/kg (12.56 μg/mL for RA)	AUC_(0__→__∞)_ = 4.10 ng/mL·h, T_max_ = 0.08 h, C_max_ = 173.19 ng/mL	[289]

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
