# Peer review of "A Comprehensive Review of Rosmarinic Acid: From Phytochemistry to Pharmacology and Its New Insight"

_molecules, 2022, doi:10.3390/molecules27103292_

Round 1
Reviewer 1 Report
The review article entitled "A comprehensive review of rosmarinic acid: from phytochemistry to pharmacology and its new insight" aims to bridge the knowledge gaps on rosmarinic acid with respect to toxicity and bioavailability for better positioning of rosmarinic acid.
The specific observations are as follows-
Introduction: The literature search strategy needs to little more elaboration.
Sources: A correlation between the genus, families, and species may be discussed to highlight the chemo-taxonomic significance.
Biosynthesis: The presence, distribution, and characteristics of the rosmarinic acid synthase enzyme should be included to emphasize its role.
Extraction from plants: Based on the available literature data the authors should draw a quantitative conclusion regarding the use of optimum solvent selection along with other conditions.
Natural derivatives in plants: Are there any derivatization patterns concerning the plant parts or families or genera?
Analytical techniques: The reported HPLC/UPLC etc methods may be tabulated along with the reported content of the RA in different plants and parts.
Clinical studies: Clinical trial registers of different countries may be searched for the recorded clinical trials as the published articles may not provide a true picture.
Application in food science: It could be application in food protection and accordingly data needs to be quantitatively assessed.
Pharmacokinetics: The review aims to highlight the toxicity and bioavailability issues but this section talks about the reported studies without any discussion on the issues.
Some aspects of reported toxicity studies are mentioned in the conclusion and future prospects section but the quantitative assessment is lacking.
Overall the manuscript's aim is not delivered by the way of data presented and discussed. It should be more than a compilation of the results reported in the literature, and it should paraphrase major findings (instead of just modifying the text from the original papers) and critically assess them. Furthermore, it should synthesize the primary reports into new conclusions, and it should provide a critical discussion of what is really known and what is missing, leading to perspectives and directions for future research in the area.
Author Response
Comment 1: Introduction: The literature search strategy needs to little more elaboration.
Response: Thank you for your comments. We have added some more elaborate search strategy in the manuscript.
Comment 2: Sources: A correlation between the genus, families, and species may be discussed to highlight the chemo-taxonomic significance.
Response: Thank you for your comments. We have added some relevant content in the manuscript.
Comment 3: Biosynthesis: The presence, distribution, and characteristics of the rosmarinic acid synthase enzyme should be included to emphasize its role.
Response: Thank you for your comments. We have added some relevant content in the manuscript.
Comment 4: Extraction from plants: Based on the available literature data the authors should draw a quantitative conclusion regarding the use of optimum solvent selection along with other conditions.
Response: Thank you for your comments. We have added some relevant content in the manuscript.
Comment 5: Natural derivatives in plants: Are there any derivatization patterns concerning the plant parts or families or genera?
Response: Thank you for your comments. We have added some relevant content in the manuscript. Additionally, to avoid superseding what really counts, we decide to move the chemical structures of the derivatives into Supplementary Figure.
Comment 6: Analytical techniques: The reported HPLC/UPLC etc methods may be tabulated along with the reported content of the RA in different plants and parts.
Response: Thank you for your comments. According to our search results, several hundreds of articles on RA determination by HPLC/UPLC have been published, in which many plants and their different parts were involved. Meanwhile, the change of a single condition (mobile phase composition, acid (or base) species or concentration in mobile phase, gradient, column, wavelength, etc.) would lead to a new method. So, if they are tabulated, it is a very huge table with a large amount of references. In addition, the emphasis of this part is quantitative method.
Comment 7: Clinical studies: Clinical trial registers of different countries may be searched for the recorded clinical trials as the published articles may not provide a true picture.
Response: Thank you for your comments. To our knowledge, there were only these two public reports about clinical trials of RA. If you can provide others, we would be grateful.
Comment 8: Application in food science: It could be application in food protection and accordingly data needs to be quantitatively assessed.
Response: Thank you for your comments. We have added some detailed data in the manuscript.
Comment 9: Pharmacokinetics: The review aims to highlight the toxicity and bioavailability issues but this section talks about the reported studies without any discussion on the issues. Some aspects of reported toxicity studies are mentioned in the conclusion and future prospects section but the quantitative assessment is lacking.
Response: Thank you for your comments. We have made modification in the manuscript and a paragraph in the last has been moved to “Pharmacokinetics”.
Comment 10: Overall the manuscript's aim is not delivered by the way of data presented and discussed. It should be more than a compilation of the results reported in the literature, and it should paraphrase major findings (instead of just modifying the text from the original papers) and critically assess them. Furthermore, it should synthesize the primary reports into new conclusions, and it should provide a critical discussion of what is really known and what is missing, leading to perspectives and directions for future research in the area.
Response: Thank you for your comments. We have made modification in the manuscript according to your suggestion and now there is a necessary discussion in each part.
Reviewer 2 Report
This is an exhaustive review concerning of rosmarinic acid. Authors characterized the sources of rosmarinic acid, extraction method, rosmarinic acid derivatives and analytical techniques. Moreover, the pharmacological activities of rosmarinic acid were widely described. Authors also showed the application of rosmarinic acid in food sciences and discussed the pharmacokinetic characters of rosmarinic acid. I have only one suggestion: Chapter 10 “Conclusions and Future Perspectives” should be more concise. Some information should be taken (line 509-520) to the previous chapters. The last paragraph it is rather a summary not conclusion and it should be taken to summary.
Author Response
Comment: Chapter 10 “Conclusions and Future Perspectives” should be more concise. Some information should be taken (line 509-520) to the previous chapters. The last paragraph it is rather a summary not conclusion and it should be taken to summary.
Response: Thank you for your comments. We have made some modifications according to your suggestions.
Reviewer 3 Report
Dear Editor,
Thank you for the opportunity to review the paper entitled "A comprehensive review of rosmarinic acid: from phytochemistry to pharmacology and its new insight". The paper is well-structured and well-written, also the used references are novel and their quantity is enough for the review type of study. Nevertheless, there are a few things that should be added/changed before the further process of publication.
Some of my overall comments for this paper:
- A new insight and trend in this area is isolation of bioactive compounds from biowaste. It would be interesting to add a paragraph about isolation of rosmarinic acid from by-products and biowaste from the circular economy perspective
- Also, Table 1 is well-structured, but for every plant there is only one reference. Including more references for the same plant would contribute to a broad review on this topic.
Author Response
Comment 1: A new insight and trend in this area is isolation of bioactive compounds from biowaste. It would be interesting to add a paragraph about isolation of rosmarinic acid from by-products and biowaste from the circular economy perspective
Response: Thank you for your comments. We have made revision according to your suggestion.
Comment 2: Also, Table 1 is well-structured, but for every plant there is only one reference. Including more references for the same plant would contribute to a broad review on this topic.
Response: Thank you for your comments. Table 1 is to show the natural plant sources of RA and to demonstrate the wide distribution among the different species in the nature. As you have mentioned, from a same plant (especially a RA-rich plant), there may be several articles on RA isolation, for example Salvia miltiorrhiza. The different articles express one thing: RA existence in this species. Additionally, there will be more references to be cited and more spaces will be occupied. So, in our opinions, for each plant there is one reference.
Round 2
Reviewer 1 Report
Since the authors have improved the manuscript as per the comments. I feel the manuscript may be accepted.
This manuscript is a resubmission of an earlier submission. The following is a list of the peer review reports and author responses from that submission.
Round 1
Reviewer 1 Report
A review on the phytochemistry, pharmacology, pharmacokinetics of rosmarinic acid, a naturally-occurring polyphenolic acid
Line 90: Table 2
- I suggest to order the table according to the used parts of the plants. E.g. plants leaves contain rosmarinic acid, etc.
- Add the author name of taxonomic identification for some species such as Adenium obesum, etc.
- Write the family names not n italic form.
- Add more data about the biological sources of RA. Based on the table data; which plant families and genera have higher number of species that contain rosmarinic acid. Which part of the plant is the most frequent source of the compound.
Line 78: Phenylpropanoid is the chemical general class of RA and similar compounds structures to RA; that have same general biosynthesis pathway. Add more about chemical data of RA.
Line 92: Explain the use of the polar solvents according to chemical structure of the compound.
Line 165: I suggest to add a table contains the important pharmacological activities of RA in short form.
Line 168; 176:” in vivo and in vitro”; write in italic form. Check across the manuscript.